# L-Shell Ionization Cross Sections for Silver by Low-Energy Electron Impacts

**Takeshi Mukoyama * and Károly Tőkési**

Institute for Nuclear Research of the Hungarian Academy of Sciences (ATOMKI), Bem tér 18/c,
H-4026 Debrecen, Hungary
\* Correspondence: hakutei@rhythm.ocn.ne.jp

**Abstract:** The L-subshell ionization cross sections and total L–X-ray production cross sections for the Ag atom by the electron impact near the ionization threshold were calculated with a classical-trajectory Monte Carlo method. The results were compared with experimental data, quantum mechanical calculations, and the cross sections by positron impact. It was demonstrated that the classical treatments are useful for electron–atom collisions at energies higher than about six times the binding energies of target electrons but overestimate L-shell ionization and L–X-ray production cross sections at low energies near the threshold. Possible reasons for this discrepancy are discussed.

**Keywords:** L-shell ionization; electron impact; silver atom; classical trajectory Monte Carlo method





## 1. Introduction

The study on inner-shell ionization processes of atoms by electron impact is of fundamental importance in atomic collisions. Reliable ionization cross sections for atomic inner shells by low-energy electrons are essential to various application fields [1,2], such as radiation physics, plasma physics, astrophysics, and medical science. They are also especially needed for techniques in materials science, for example, electron-probe microanalysis (EPMA), auger-electron spectroscopy (AES), and electron energy-loss spectroscopy (EELS).

In addition, it is interesting to compare the cross sections by electrons with those by positrons to elucidate the differences in interactions with the matter between particles and antiparticles. At high energies, ionization cross sections by positrons are almost equal to those by electrons, but there is a large difference for the low-energy region near the ionization threshold.

A number of theoretical models to calculate inner-shell ionization cross sections by electron impacts have been developed. It is well known that at high incident energies the plane-wave Born approximation (PWBA) can give the correct asymptotic behavior of the ionization cross sections. However, for the low-energy region near to the ionization threshold, the experimental cross sections deviate from the PWBA considerably. This is mainly ascribed to a distortion of the incident electron trajectory caused by the Coulomb field of the target nucleus. Several methods to modify the PWBA have been proposed [3,4] and used to compare with the experimental data. To date, many other theoretical models, such as the classical binary-encounter dipole model [5] and the distorted-wave Born approximation (DWBA) [6–8], have been developed and applied to analyze the experimental results.

The experimental values for K-shell ionization cross sections were compiled by Long et al. [9] and Llovet et al. [10] and the measured K-, L-, and M-shell ionization cross sections were compared with various theoretical calculations. Patoary et al. [11] presented a comparison of measured L-subshell and total L–X-ray production cross sections with their semi-empirical models. Many experimental data for electron impacts have been reported on, but most of them are concentrated on K-shell ionization cross sections. In the case of K-shell ionization, although a slight discrepancy is found for the values from different

authors, the experimental data are in agreement with the theoretical calculations with each other.

On the other hand, for the L-shell ionization cross sections or L–X-ray production cross sections, the number of measurements is small, and the agreement between the theory and experiment is not satisfactory. In addition, most experimental studies are for total L–X-ray production cross sections and the experimental data for separate subshell ionization cross sections are rather scarce. One of the reasons for experimental difficulties consists of the fact that vacancies produced by electron impacts can transfer to other subshells, and subshell ionization cross sections depend on atomic relaxation parameters, such as subshell fluorescence yields and Coster–Kronig transition probabilities between subshells.

In the present work, we calculated the L-subshell ionization cross sections of the Ag atom by electron impact by the use of the classical trajectory Monte Carlo method (CTMC) [12]. Recently we have shown that the CTMC method can well predict L-shell ionization cross sections by positron impact [13]. In the present work, we applied a similar approach to the case of electron impact ionization and calculated the L-shell ionization cross sections for Ag.

For Ag, Reusch et al. [14] measured L-subshell ionization cross sections by electron impacts in the energy range between 50 and 200 keV. Their experimental values are in good agreement with the relativistic PWBA (RPWBA) calculations of Scofield [15]. Therefore, we are interested in the lower-energy region and studied L-shell ionization cross sections by electron impact with incident energy less than 30 keV.

In this energy region, one measurement of subshell ionization cross sections and three experimental studies on the total L-shell X-ray production cross sections have been reported. Sepúlveda et al. [16] measured L-subshell ionization cross sections for Ag between 6 and 25 keV. The total L–X-ray production cross sections were observed by two groups between 5 and 30 keV [17–19]. The calculated values obtained with the CTMC method were compared with these experimental results and with quantum mechanical calculations. A comparison with the corresponding theoretical data by positron impact was also made.

## 2. Theoretical Model

The CTMC method is a non-perturbative method based on classical dynamics and all the interactions between the colliding particles are automatically taken into consideration [20]. In order to apply the CTMC to atoms other than hydrogen, we used the screened hydrogenic model and the screening constant was determined according to Slater's rule [21]. Then the system could be treated as the three-body system, which consists of the incident electron, the target nucleus, and one L-shell electron. They are assumed to interact with each other through the pure Coulomb field.

Figure 1 shows the relative position vectors of particles in the present three-body system. The projectile electron $P$ is moving along with the velocity $\vec{v}_p$. The symbol $O$ indicates the center of mass of the target system. The target nucleus is located at $T$ and the electron is at $e$. The impact parameter of the projectile is indicated as $b$.

In general, the classical equations of motion by Newton for a three-body system are solved numerically for a large number of trajectories with given initial conditions. The initial conditions correspond to the impact parameter of the incident electron, the position, and the velocity of the target electron. For each trajectory, these parameters are selected randomly by the use of pseudo-random numbers.

With each initial condition determined, the equations of motion for the system are solved by numerical integration with respect to time using the standard Runge–Kutta method. In order to distinguish the various final states, the exit channels were tested at large distances from the collision center.

The total ionization cross section was obtained from

$$\sigma = \frac{2\pi \, b_{\max} \sum_j b_j^{(i)}}{N} \, ,$$

(1)

where $N$ is the total number of trajectories with impact parameters less than $b_{max}$ and $b_j^{(i)}$ is the actual impact parameter of the $j$-th trajectory corresponding to the ionization channel. The standard deviation of the ionization cross section is expressed as

$$\Delta\sigma = \sigma \left( \frac{N - N^{(i)}}{N\,N^{(i)}} \right)^{1/2} , \qquad (2)$$

where $N^{(i)}$ is the number of trajectories that satisfies the criterion for the ionization process.

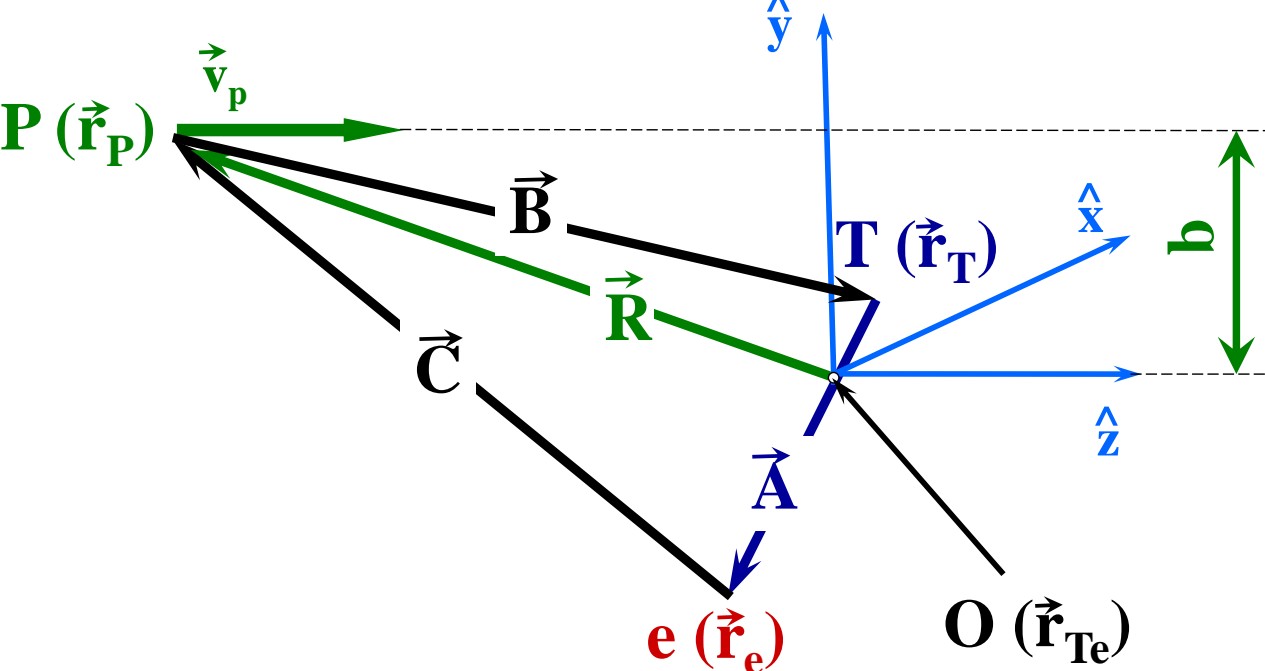

**Figure 1.** Relative positions of the three-body system used for the CTMC. The symbol $P$ represents the projectile electron, $O$ the center of mass of the target system, $T$ the nucleus of the target atom, $e$ the target electron, and $b$ the impact parameter.

### 3. Results and Discussion

The L-subshell ionization cross sections of Ag were calculated by the CTMC for incident electrons with kinetic energies below 30 keV. In order to achieve good statistical accuracy, we needed 10 million histories for $N$ in Equation (1) for each incident energy. The binding energy of the target electron was taken from the table of Bearden and Burr [22] to be 3806 eV for the $L_1$ shell, 3524 eV for the $L_2$ shell, and 3351 eV for the $L_3$ shell.

Figure 2 shows the $L_1$-shell ionization cross sections of Ag as a function of the incident electron energy. The CTMC calculations were compared with experimental values of Sepúlveda et al. [16] and the DWBA calculations of Bote et al. [23]. Experimentally the partial L–X-ray production cross sections were determined for L–X-ray components, such as $L_\alpha$, $L_\beta$, and $L_\gamma$ lines, from the measured L–X-ray spectra. Then they were converted to the L-subshell ionization cross sections by the use of fluorescence yields and Coster–Kronig coefficients. The different choices of these atomic parameters give different L-subshell ionization cross sections. Sepúlveda et al. [16] used two sets of parameters, i.e., Perkins et al. [24] and Campbell [25], and obtained two different sets of the experimental values. The experimental cross sections with the parameters of Perkins et al. [24] are shown as Exp-a and those with the values of Campbell [25] are represented as Exp-b in the figure.

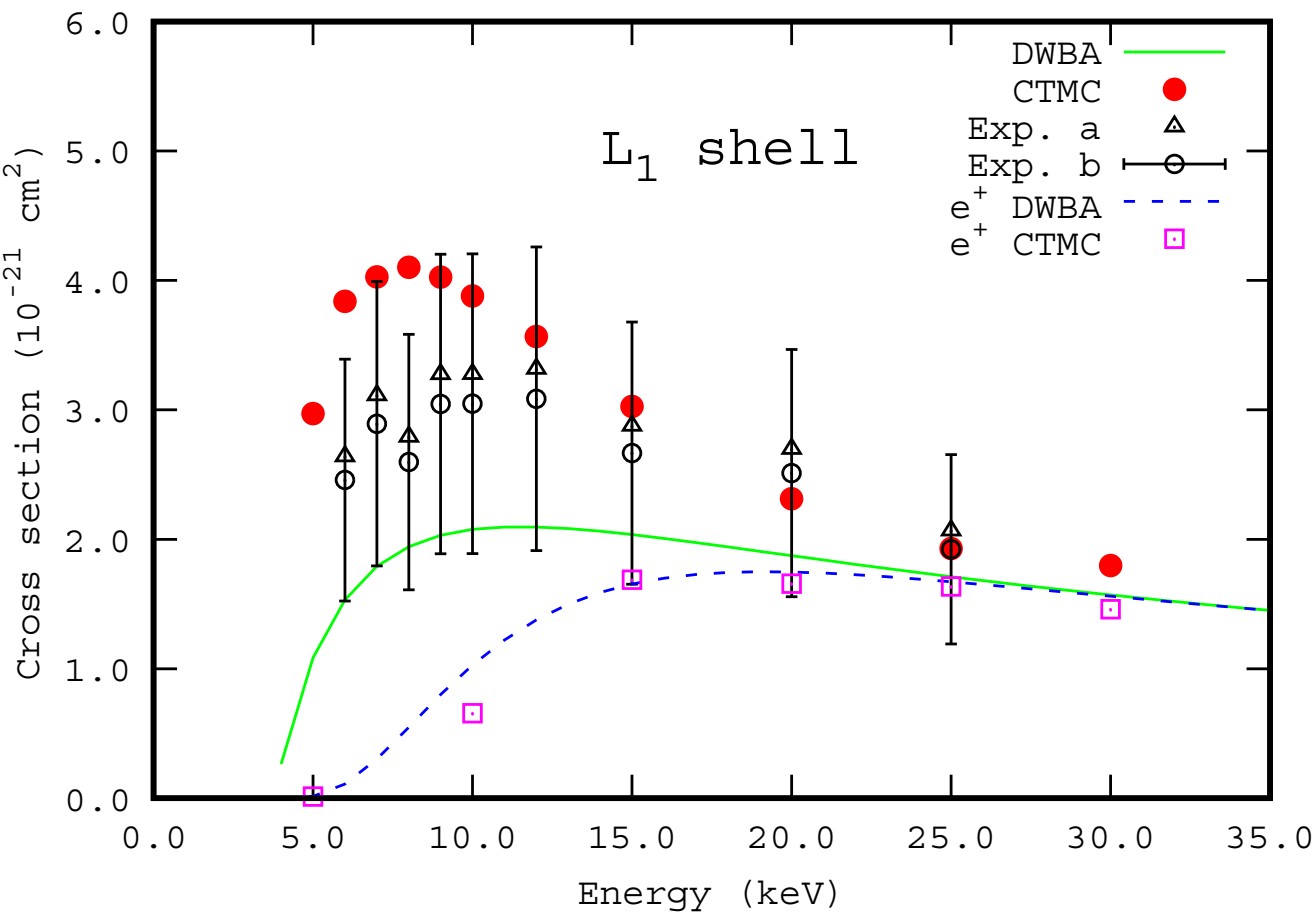

**Figure 2.** $L_1$–shell ionization cross sections of Ag by electron impact as a function of incident energy. The solid circles are the present CTMC calculations, the solid lines are the DWBA values of Bote et al., the open triangles the experimental data of Sepúlveda et al. [16] with parameters of Perkins et al. [24], and the open circles the same experimental data [16] with parameters of Campbell [25]. For comparison, the theoretical results by positron impact with the DWBA are shown by the dashed curve and the CTMC values are plotted by the open squares.

The present results are about 30% larger than the experimental values between 5 and 10 keV, but agree with the experiment within the experimental error for higher energies. The DWBA values are smaller than the CTMC values and the experimental data in the low-energy region. However, all the values are in good agreement with each other for energies higher than 25 keV.

For comparison, the CTMC [13] and the DWBA [23] calculations for positrons are also plotted in the figure. Both cross sections by positron impacts were smaller than those by electrons in the low-energy region due to the Coulomb repulsion for the incident positron, but become almost the same in the high-energy region. In the case of positron impact, the CTMC results were in good agreement with the DWBA calculations.

Figures 3 and 4 show a comparison of the theoretical and experimental cross sections for $L_2$- and $L_3$-shell ionization, respectively. It is clear that the general trends of both cross sections as functions of impact energy are similar to those for the $L_1$-shell ionization cross section. For the $L_2$ shell, the CTMC overpredicts the ionization cross section by about 40% at 7 keV, while the $L_3$-shell ionization cross sections are 30% higher at the same energy. All the theoretical values, including those by positrons, are in agreement with each other for energies higher than 20 keV. On the other hand, the experimental values are systematically larger than the theoretical ones.

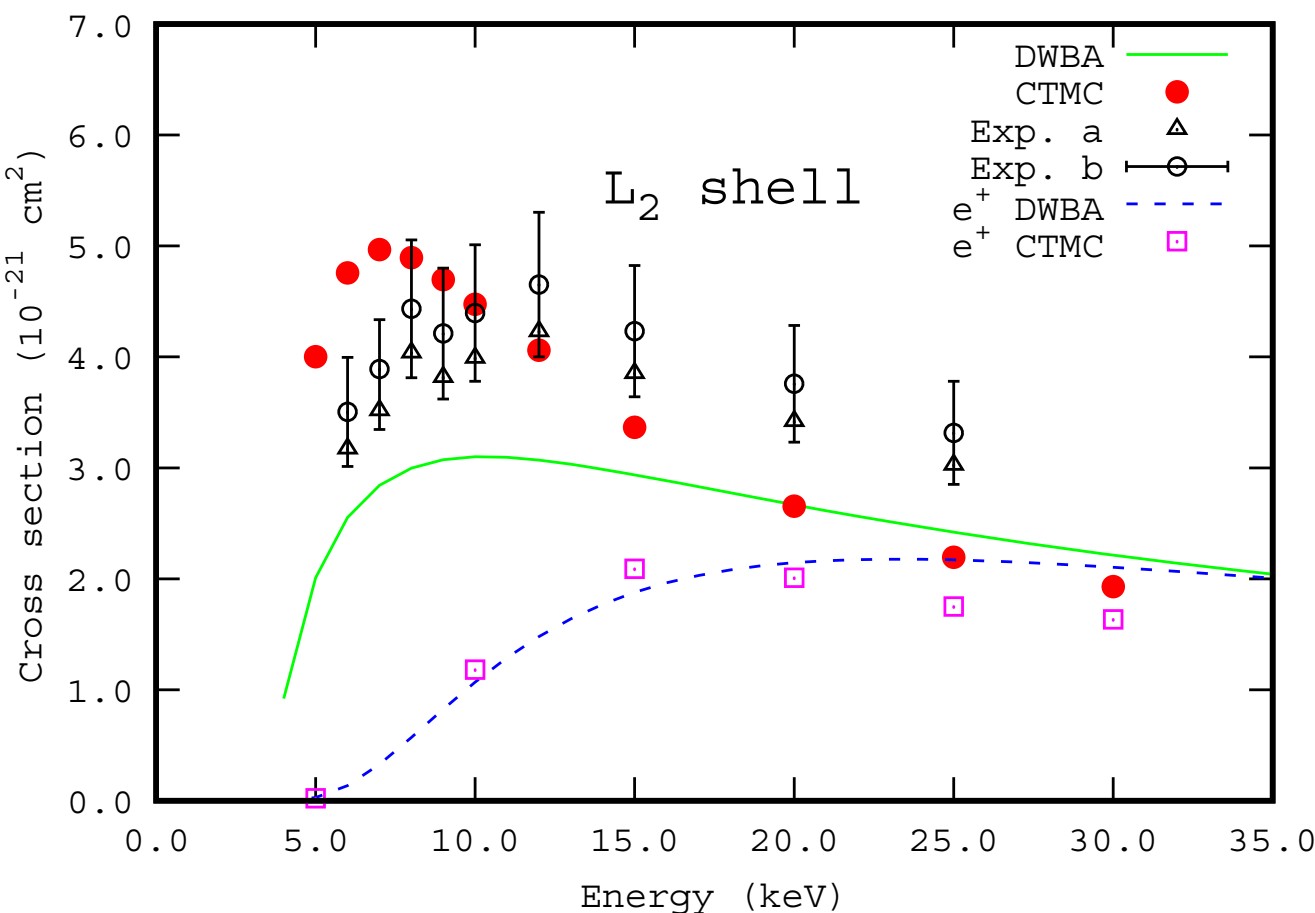

**Figure 3.** $L_2$–shell ionization cross sections of Ag by electron impacts as functions of incident energy. The solid circles are the present CTMC calculations, the solid lines represent the DWBA values of Bote et al., the open triangles represent the experimental data of Sepúlveda et al. [16] with parameters of Perkins et al. [24], and the open circles represent the same experimental data [16] with parameters of Campbell [25]. For comparison, the theoretical results by positron impacted with the DWBA are shown by the dashed curve and the CTMC values are plotted by the open squares. .

In order to compare the present results with the measured values for total L–X-ray production cross sections, the calculated $L_i$-subshell ionization cross sections $\sigma_i$ have to be converted into the total L–X-ray production cross section $\sigma_L^X$:

$$\sigma_L^X = \sum_{i=1}^{3} \sigma_i^T \omega_i \,, \tag{3}$$

where

$$\sigma_i^T = \sigma_i + \sum_{j<i} \sigma_j^T f_{ji} \,, \tag{4}$$

is the total $L_i$-subshell vacancy production cross section, $\omega_i$ is the $L_i$-subshell fluorescence yield, and $f_{ji}$ is the Coster–Kronig transition probability from the $L_j$ subshell to the $L_i$ subshell.

In the present work, we used two different data sets of atomic parameters, $\omega_i$ and $f_{ji}$. Set a is taken from Perkins et al. [24] and data set b is from Campbell [25]. This choice was the same for Sepúlveda et al. [16]. The total L–X-ray production cross sections corresponding to the CTMC and the DWBA were calculated with these two parameter sets from the subshell ionization cross sections.

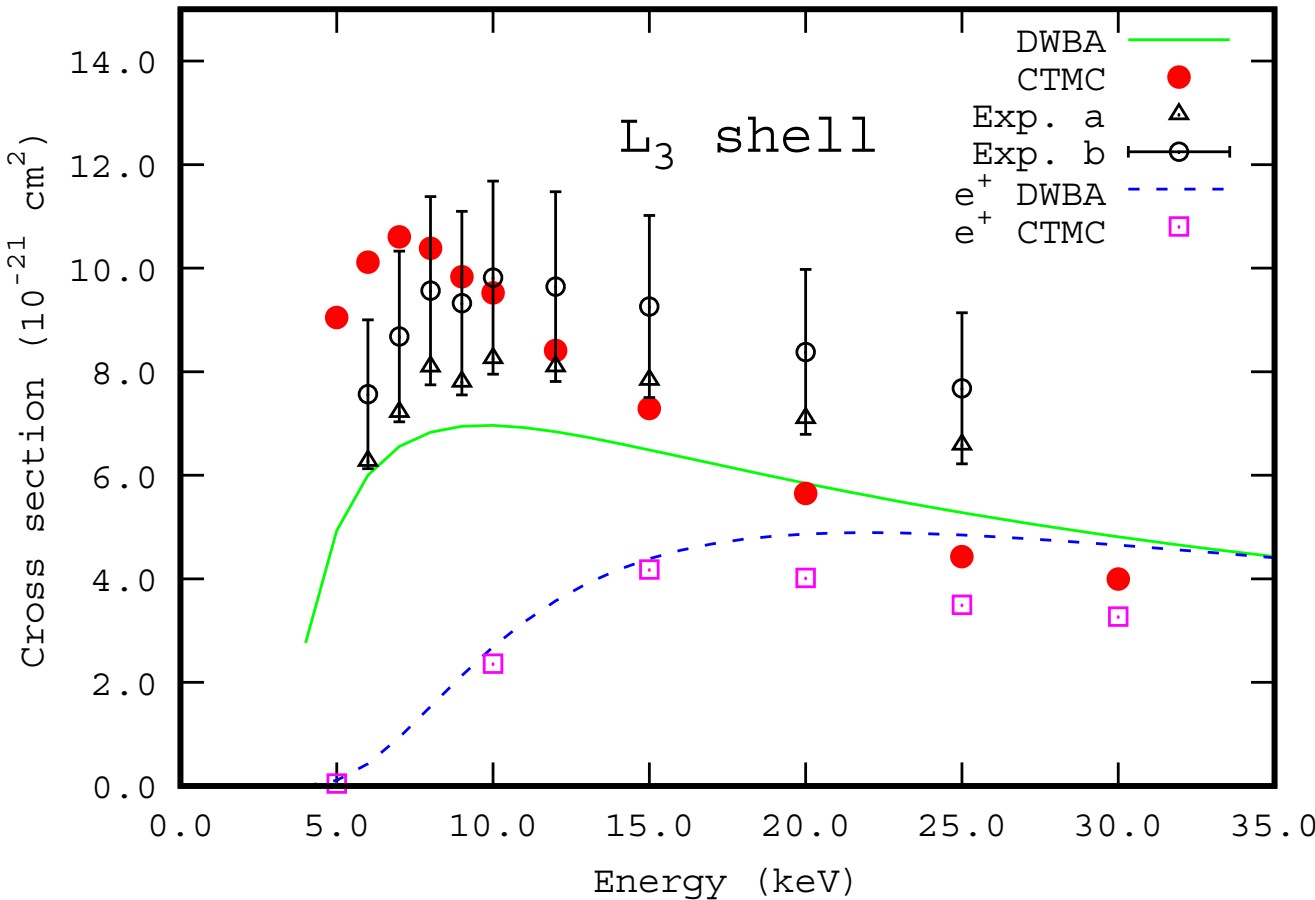

**Figure 4.** $L_3$–shell ionization cross sections of Ag by electron impacts as functions of incident energy. The solid circles are the present CTMC calculations, the solid lines represent the DWBA values of Bote et al., the open triangles represent the experimental data of Sepúlveda et al. [16] with parameters of Perkins et al. [24], and the open circles represent the same experimental data [16] with parameters of Campbell [25]. For comparison, the theoretical results of the positron impact with DWBA are shown by the dashed curve and the CTMC values are plotted by the open squares.

Figure 5 compares the present CTMC results of the total L–X-ray production cross sections with the experimental data of Wu et al. [17], Sepúlveda et al. [16], and Zhao et al. [19] as well as the theoretical DWBA values [23]. In the case of the experimental data of Sepúlveda et al. [16], only the cross sections with parameter set a were plotted in the figure, because the effect of a different choice of atomic parameters was small.

It can be seen from the figure that the CTMC values with both sets of parameters are in agreement with the DWBA values and the experimental data of Wu et al. [17] and Zhao et al. [19] for the energy region higher than 20 keV. However, when the incident energy is near the ionization threshold, the CTMC overpredicts all other theoretical and experimental values. This trend is similar to the case of L-subshell ionization cross sections.

The experimental results of Sepúlveda et al. [16] are larger than other experimental data. On the other hand, the values of Wu et al. [17] are systematically smaller. The values of Zhao et al. [19] are between the other two values. The reasons for this discrepancy were discussed by Sepúlveda et al. [16] and Zhao et al. [19] and may be ascribed to the target thickness, calibration of detection efficiency, corrections for multiple scattering and backscattering of electrons, and the size of the incident electron beam spot.

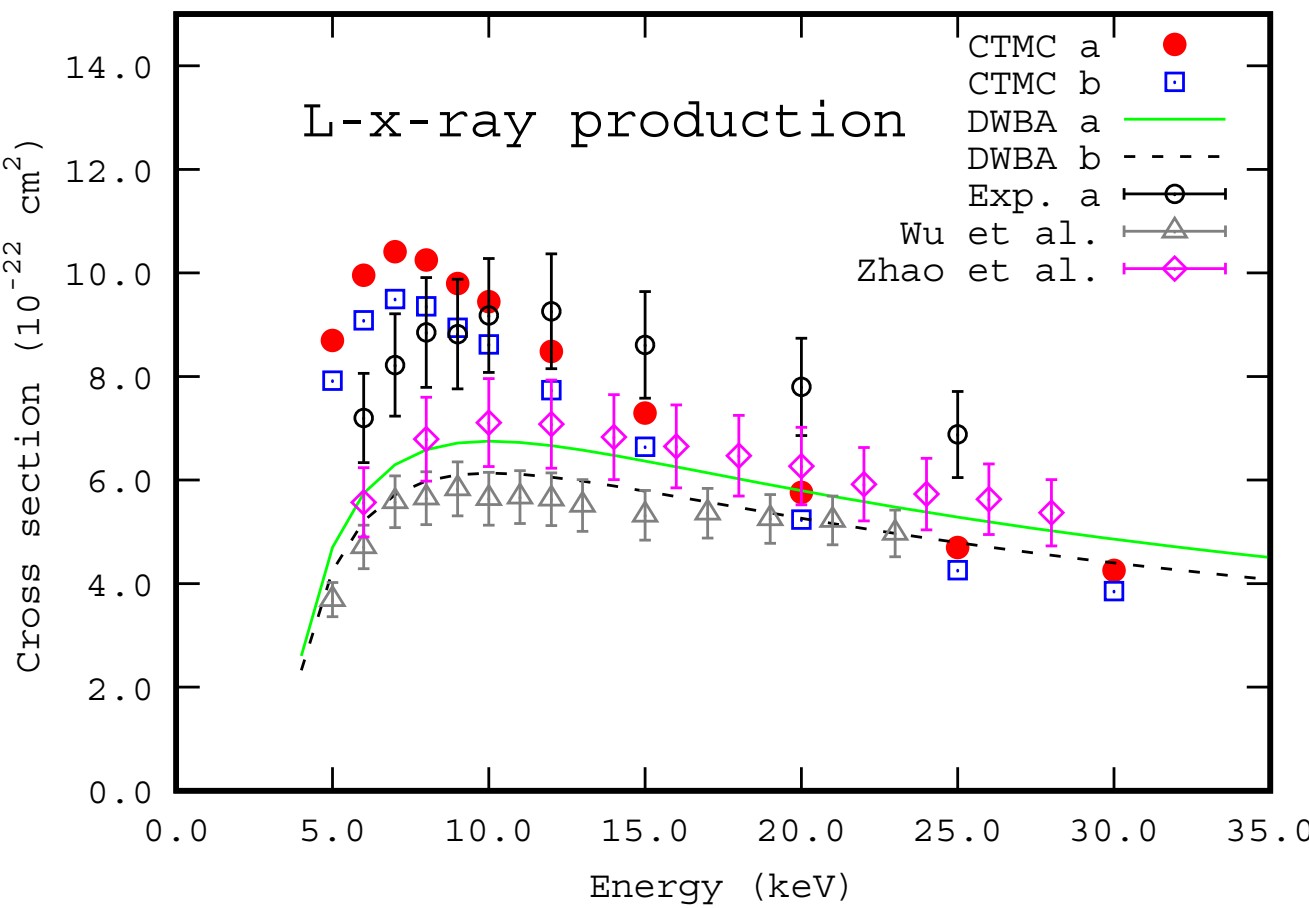

**Figure 5.** Total L–X-ray production cross sections of Ag by the electron impact as a function of incident energy. The solid circles are the CTMC with parameter set a by Perkins et al. [24], the open squares are the CTMC with set b by Campbell [25], the solid line the DWBA [23] with set a, the dashed line the DWBA [23] with set b, the open circles are the experimental data of Sepúlveda et al. [16] with set a, the open triangles are the experimental data of Wu et al. [17], and the open diamonds the experimental data of Zhao et al. [19].

It can be seen from Figures 2–5 that the CTMC gives larger cross sections than the DWBA in the energy region near the ionization threshold. However, both values are in good agreement for incident energies higher than 20 keV, which is about six times the binding energy of the target electron. It is also interesting to note that for positron impact ionization, the CTMC results agree well with those of the DWBA, as can be seen in Figures 2–4. Hippler and Jitschin [26] pointed out that the electron exchange effect between the incident and target electrons, which is not included in the classical model, such as the CTMC, is important at low incident energies. These facts suggest that the possible reason for the discrepancy between the CTMC and the DWBA is the electron exchange effect. Prasad [27] showed that the exchange effect reduces the L-subshell ionization cross section by electron impact considerably in the energy region near the ionization threshold. If we take into consideration the electron exchange effect, the CTMC values decrease in the low-energy region and becomes closer to the DWBA and the experimental data.

## 4. Conclusions

We calculated the L-shell ionization cross sections for Ag by electron impacts with the CTMC method. The target atom was considered in the screened hydrogenic model. The CTMC calculations were performed for the three-body system by producing a large number of trajectories.

In the energy region higher than 20 keV, both calculated L-subshell ionization cross sections and L-shell X-ray production cross sections agreed with the DWBA values and the experimental data. When the incident energy became lower and was near the maximum of the cross sections, the CTMC overpredicted L-shell ionization cross sections. On the other hand, in the case of positron impacts, the CTMC was in good agreement with the DWBA. This fact suggests that the electron exchange effect plays an important role in the low-energy region.

There are systematical discrepancies between experimental L–X-ray production cross sections in the low-energy region and only data from one experiment have been reported for L-subshell ionization cross sections. More experimental studies on L-subshell ionization cross sections for Ag with low-energy electrons are needed.

**Author Contributions:** Both authors discussed the results and contributed to the final manuscrpt. All authors have read and agreed to the published version of the manuscript.

**Funding:** This research received no external funding.

**Institutional Review Board Statement:** Not applicable.

**Informed Consent Statement:** Not applicable.

**Data Availability Statement:** Correspondence and requests for the data should be addressed to T.M.

**Conflicts of Interest:** The authors declare no conflict of interest.

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
