# Peer review of "L-Shell Ionization Cross Sections for Silver by Low-Energy Electron Impacts"

_atoms, doi:10.3390/atoms10040116_

Round 1

Reviewer 1 Report

This article reports on a classical-trajectory Monte Carlo calculation of L-subshell and total L ionization cross sections by low-energy electrons, for Ag atoms. The obtained results are compared with other calculations and experiments. 

The used model is well explained and the discussion of the results and conclusions are acceptable, in my opinion. 

I recommend that the paper should be accepted after a few corrections. 

There are a few issues that should be addressed: 

  1. The captions of Figs. 2, 3, and 4, mention “For comparison, the theoretical results by positron impact with the DWBA are shown by the dashed curve and the CTMC values are plotted by the open squares.” 

However, neither dashed curve nor opens squares are visible in the mentioned figures, at least in the version of the article I had access to. Please correct. 

  1. Some corrections of the text: 

  1. On page 4, 2nd paragraph, 2nd line: please remove the dot in “region.by” 

  1. On page 6, 4th paragraph, 4th line: please replace “becomes” by “become”. 

  1. On page 7, 4th paragraph, 1rst line: please replace “Figure 5 indicates comparison of the“ by “Figure 5 indicates compares the“. 

  1. On page 7, 5th paragraph, please replace “approaches near to the” by “approaches the”. 

  1. On page 8, 2nd paragraph, 4th line: please replace “6 times of the binding energy” by “6 times the binding energy”. 

  1. On page 8, 2nd paragraph, 13th please replace “near to the ionization” by “near the ionization”.

Author Response

Thank you for your commets.  I am sorry we have made mistake in Figs. 2, 3, and 4.  We replaced them by the correct ones which include the positron data.  For other comments, we followed the comments and modified our manuscript.

Reviewer 2 Report

Reviewer's comments

Manuscript ID: atoms-1934253
Type of manuscript: Article
Title: L-shell ionization cross sections for silver by low-energy electron
impact
Authors: Karoly Tokesi, Takeshi Mukoyama
Interaction of Electrons with Atoms, Molecules and Surfaces
https://www.mdpi.com/journal/atoms/special_issues/Interaction_Electrons

Authors have calculated the L-shell ionization cross sections and total L-x-ray
production cross sections for Ag atom by low-energy electron impact with the
classical-trajectory Monte Carlo (CTMC) method. The target atom is considered in
the screened hydrogenic model and the CTMC calculations were performed for
three-body system. The results are compared with the experimental data, the
the distorted-wave Born approximation (DWBA) calculations and the cross sections
by positron impact. Authors demonstrated that the classical treatments
are working well for electron-atom collisions at energies higher than about 6
times of the binding energies of target electrons, but in case at low energies
near to the threshold overestimate L-shell ionization and L-x-ray production
cross sections. In the case of positron authors showed that impact the CTMC is
in good agreement with the DWBA calculations. Authors conclude that the electron
exchange effect plays an important role in the low-energy region.

I believe that this paper is interesting and useful for the readers in
the field. I would like to recommend publication of this paper in journal Atoms.

A few typo errors:

Page 4, line 11 from the top

It stands:
... in low-energy region.by the use of...
It should be:
... in low-energy region by the use of...

Page 9, References

It stands:
[1] C. J. Powell, Rev. Mod. Phys. 1976, 48,33.
It should be:
[1] C. J. Powell, Rev. Mod. Phys. 1976, 48, 33.

Page 10, References
It stands:
[16] A. Sepúlveda, A. P. Bertol, M. A. Z. Vasconcellos, J. Trincavelli, R.
Hinrichs, and G. Castellano, J. Phys. B: At. Mol. Opt. Phys. 2014, 47,
215006
It should be:
[16] A. Sepúlveda, A. P. Bertol, M. A. Z. Vasconcellos, J. Trincavelli, R.
Hinrichs, and G. Castellano, J. Phys. B: At. Mol. Opt. Phys. 2014, 47,
215006.

Page 10, References
It stands:
[23] D. Bote, F. Salvat, A. Jablonski, and C. J. Powell, At. Data Nucl. Data
Tables 2009, 95, 871 .
It should be:
[23] D. Bote, F. Salvat, A. Jablonski, and C. J. Powell, At. Data Nucl. Data
Tables 2009, 95, 871.

Author Response

Thank you for your comments.  According to them, we modified our manuscript.

Reviewer 3 Report

Mukoyama and Tökesi calculate L-subshell ionization cross section of silver atoms by electron impact close to the ionization threshold and compare their results with previous calculations and experimental data. In addition, the total X-ray production cross section was obtained and gain compared with results from the literature.

The authors claim that they show also data of positron impact, however, in all figures I cannot see them. This has to be corrected.

The terminus low-energy electron impact might be somewhat irritating since the threshold for L-shell ionization of silver is above 3.5 keV and many reader associate with low-energy electrons energies below 10eV.

X-ray is typically written with a capital X.

Besides this, I made some 80+ annotations to the pdf file the authos might consider when revising the manuscript.

I do not have to see the paper again and recommend publication after fixing the points mentioned above.

Author Response

Thank you for your comments and suggestions.  For Figs. 2, 3, and 4, we are sorry to have made mistakes.  We replaced them by the correct ones, which include the positron data.

The words "low-energy electrons" are sometimes troublesome, becasue what is "low" is different in different fields.  In the case of inner-shell ionization, we usually use "the low-energy region" when the ionization cross section deviates from the simple first Born approximation.  The reviewer suggested the title "Near threshold", but we are not so much interested in near-threshold structure.  We want to keep the title in the original manuscript.  Except for this, we followed all the comments and suggestions of the reviewers and modifed the manuscript.